# The Development of a Risk Assessment Model for Inedible Rendering Plants in Canada: Identifying and Selecting Feed Safety-Related Factors

**DOI:** 10.3390/ani14071020

**Published:** 2024-03-27

**Authors:** Virginie Lachapelle, Geneviève Comeau, Sylvain Quessy, Romina Zanabria, Mohamed Rhouma, Tony van Vonderen, Philip Snelgrove, Djillali Kashi, My-Lien Bosch, John Smillie, Rick Holley, Egan Brockhoff, Marcio Costa, Marie-Lou Gaucher, Younes Chorfi, Manon Racicot

**Affiliations:** 1Canadian Food Inspection Agency, 3200 Sicotte, St-Hyacinthe, QC J2S 2M2, Canada; genevieve.comeau@inspection.gc.ca (G.C.); manon.racicot@inspection.gc.ca (M.R.); 2Faculty of Veterinary Medicine, Université de Montréal, 3200 Sicotte, St-Hyacinthe, QC J2S 2M2, Canada; sylvain.quessy@umontreal.ca (S.Q.); mohamed.rhouma@umontreal.ca (M.R.); marcio.costa@umontreal.ca (M.C.); marie-lou.gaucher@umontreal.ca (M.-L.G.); younes.chorfi@umontreal.ca (Y.C.); 3Canadian Food Inspection Agency, 1400 Merivale, Ottawa, ON K1A 0Y9, Canada; romina.zanabriaeyzaguirre@inspection.gc.ca; 4Canadian Food Inspection Agency, 59 Camelot Drive, Ottawa, ON K1A 0Y9, Canada; tvanvond@gmail.com (T.v.V.); philip.snelgrove@inspection.gc.ca (P.S.); 5Sanimax, 2001 Av. de La Rotonde, Lévis, QC G6X 2L9, Canada; djillali.kashi@sanimax.com; 6Animal Nutrition Association of Canada, 300 Sparks St., Suite 1301, Ottawa, ON K1R 7S3, Canada; boschmylien@gmail.com; 7College of Agriculture and Bioresources, University of Saskatchewan, Agriculture Building 51 Campus Drive, Saskatoon, SK S7N 5A8, Canada; johnfsmillie@gmail.com; 8Department of Food and Human Nutritional Sciences, University of Manitoba, Winnipeg, MB R3T 2N2, Canada; rick.holley@umanitoba.ca; 9Canadian Pork Council, 900-220 Laurier Ave. W., Ottawa, ON K1P 5Z9, Canada; brockhoff@cpc-ccp.com

**Keywords:** feed safety, risk assessment, rendering plant, livestock feed, risk-informed oversight

## Abstract

**Simple Summary:**

In 2019, the Canadian Food Inspection Agency started the development of a risk assessment model for regulated inedible rendering plants in Canada that manufacture ingredients for use in livestock feed. The model is designed to assess the level of risk of each rendering plant to the health of livestock consuming the products used as feed as well as the health of humans eating food products derived from these animals. The output of the model will inform the risk-based allocation of the agency’s inspection resources. The aim of the present study was to identify and select factors associated with rendering plant activities to be included in the risk assessment model based on the scientific literature and expert advice. A final list of 32 factors assessed by 179 evaluation criteria according to rendering practices was selected for inclusion in the model and will be used for the next development step, which will estimate the relative contribution of each criterion to the overall risk of a rendering plant. Once completed, this model will provide a systematic and evidence-based approach to help manage the animal and human health risks in the rendering sector.

**Abstract:**

The Canadian Food Inspection Agency (CFIA) is developing an establishment-based risk assessment model to categorize rendering plants that produce livestock feed ingredients (ERA-Renderer model) according to animal and human health risks (i.e., feed safety risks) and help in determining the allocation of inspection resources based on risk. The aim of the present study was to identify and select feed-safety-related factors and assessment criteria for inclusion in the ERA-Renderer model. First, a literature review was performed to identify evidence-based factors that impact the feed safety risk of livestock feed during its rendering processes. Secondly, a refinement process was applied to retain only those that met the inclusion conditions, such as data availability, lack of ambiguity, and measurability. Finally, an expert panel helped in selecting factors and assessment criteria based on their knowledge and experience in the rendering industry. A final list of 32 factors was developed, of which 4 pertained to the inherent risk of a rendering plant, 8 were related to risk mitigation strategies, and 20 referred to the regulatory compliance of a rendering plant. A total of 179 criteria were defined to assess factors based on practices in the Canadian rendering industry. The results of this study will be used in the next step of the model development to estimate the relative risks of the assessment criteria considering their impact on feed safety. Once implemented, the CFIA’s ERA-Renderer model will provide an evidence-based, standardized, and transparent approach to help manage the feed safety risks in Canada’s rendering sector.

## 1. Introduction

According to Statistics Canada, between 2021 and 2023, around 12.1 million cattle, 13.8 million hogs, 114 million poultry (chickens and turkeys), and over 166 000 tons of fish were reared or caught, slaughtered, and processed for human consumption in Canada annually [1,2,3,4]. However, it is roughly estimated that around 50% of the live weight of each slaughtered or processed food animal is not eaten by consumers (e.g., bones, fats, blood, feathers, viscera) [5]. Deceased animals on farms, as well as scraps, fats, bones, spoiled meat, and used cooking oils from all types of food establishments, such as grocery stores and restaurants, also enter the inedible production chain. Approximately 2.2 million tons of inedible animal by-products are generated in Canada annually and are subjected to rendering processes [6]. Considered “the original recycling”, rendering has been applied for centuries, although the production of rendered products for use as feed ingredients only started in North America in the early 1900s [7]. Rendering operations generally consist of the cooking and drying of meat and/or other animal by-products not used for human consumption to recover fats and proteins [6]. These substances are then transformed into various useful products, including feed ingredients such as meals (bone, blood, feather, fish), tallow and poultry fat, biodiesel, and fertilizers. In Canada, a mammalian-to-ruminant-feed ban was enacted in 1997, prohibiting proteins derived from most mammals (including ruminants) from being fed to ruminants, with the intent of limiting the dissemination of the prion-causing Bovine Spongiform Encephalopathy (BSE). In accordance with the Health of Animals Regulations, Sections 162 to 164, such prohibited material are defined as anything that is or that contains any protein originating from a mammal, other than the following: a porcine or equine; milk or products of milk; gelatin or products of gelatin derived exclusively from hides or skins; blood or products of blood; or ruminant rendered fats containing no more than 0.15% insoluble impurities or their products [8]. In 2007, an enhanced feed ban was legislated to remove specified risk materials (SRMs), comprising the skull, brain, trigeminal ganglia, eyes, tonsils, spinal cord, and dorsal root ganglia of cattle aged 30 months or older and the distal ileum of cattle of all ages, containing higher concentrations of the BSE prion, from animal feed, pet food, and fertilizers [9]. The removal of SRMs from these products, combined with their exclusion from the human food supply since 2003, allowed for the increased protection of animal and human health and accelerated the eradication of BSE in Canada [10]. It also strengthened the regulatory oversight of rendering plants, with a focus on facilities handling prohibited material and/or SRMs.

The Canadian Food Inspection Agency (CFIA) is responsible for enforcing federal laws and regulations in regard to safeguarding food, animals, and plants, including the safety of livestock feed under the Feeds Act and Regulations as well as the Health of Animals Act and Regulations [8,11,12,13]. The CFIA is thus responsible for the oversight of a variety of feed establishments, such as commercial mixed feed manufacturers (including premix facilities), on-farm feed mills, single-ingredient feed manufacturers, rendering plants, retail outlets, and feed conveyances. In recent years, the CFIA has initiated the modernization of its inspection system to implement a risk-informed approach for the supervision of the whole procedure. In this context, in 2018, the CFIA initiated the development of an establishment-based risk assessment (ERA) model applicable to Canadian feed mills under the agency’s jurisdiction (ERA-Feed Mill model) to assess the animal health and food safety risks associated with livestock feed products (i.e., feed safety risks) [14,15]. Subsequently, in 2019, the CFIA began the development of an ERA model applicable to single-ingredient feed manufacturers, targeting all inedible rendering plants that are required to obtain a permit to operate under the authority of the Health of Animals Regulations: the ERA-Renderer model. For the purpose of this risk assessment model, rendering operations involve the processing of raw animal by-products and/or recycled fats from animals to produce protein meals and fats, such as meat and bone meals, blood meals, feather meals, fish meals, and tallow and poultry fats, for use in livestock feed only. As defined in the Feeds Act [11], livestock refers to animals such as those of the bovine, caprine, equine, ovine, poultry, porcine, and fish species and excludes all traditional pet species. To follow the same approach as the ERA-Feed Mill model, the ERA-Renderer model will consider the risk to the health of livestock that consume the rendered products used as feed ingredients, as well as the risk to the health of humans eating food products derived from these animals. Hence, the model covers both animal health and food safety concerns related to feed. In Canada, between 40 and 45 inedible rendering plants request a permit every year and are typically distributed in four areas of the country (West, Ontario, Quebec, and Atlantic). Although the current oversight by the CFIA in rendering plants uses a risk-based approach that considers the BSE risk (i.e., handling of prohibited material and SRMs), the ERA-Renderer model will deliver an additional layer of risk information, providing the agency with a more comprehensive, evidence-based, and systematic risk assessment of Canadian rendering plants for all currently known feed safety risks. To support the development of this tool, the present study was conducted to identify and select, according to a literature review and expert advice, feed safety-related factors to be included in the ERA-Renderer model and to define criteria assessing each of the selected factors.

## 2. Materials and Methods

The development of the ERA-Renderer model was initiated in December 2019 by a technical working group (TWG) composed of nine professionals from the CFIA (three veterinarians, one statistician, one food safety specialist, and four feed program/operations specialists) as well as five researchers from the Université de Montréal. The TWG established the research questions in relation to the scope of the model and conducted the literature search to identify factors to be included in the model. In March 2020, a scientific advisory committee (SAC) composed of 7 Canadian experts was created to provide expert advice while reviewing and finalizing the selection of factors and to support the scientific development of the model in the following steps. The SAC members were selected based on their feed and food safety expertise and their experience in the Canadian rendering industry. Members of the SAC worked in various sectors, including the livestock feed and/or rendering industry [*n* = 3], academia (Université de Montréal [*n* = 1], University of Manitoba [*n* = 1], and University of Saskatchewan [*n* = 1]), and the government (CFIA [*n* = 1]). The SAC was consulted regularly via email, conference calls, and meetings from 2020 to 2021. 

The approach used in the current study to identify and select factors for inclusion in the ERA-Renderer model was similar to the methodology followed by the CFIA to develop other risk assessment models [14,16,17]. The research query in the context of the ERA-Renderer model development was to identify, from the scientific literature, what factors pertaining to inedible rendering plant practices would be linked to either an increase or a decrease in feed safety risks. To respond to this inquiry, a 5-step literature review was performed by the TWG between January and March 2020 to retrieve the scientific literature of interest. In brief, a search was conducted in the online databases PubMed (MEDLINE; https://pubmed.ncbi.nlm.nih.gov/, accessed on 14 March 2024 and PubMed Central; https://www.ncbi.nlm.nih.gov/pmc/, accessed on 14 March 2024), CAB abstracts (https://www.cabidigitallibrary.org/product/ca, accessed on 14 March 2024), and Global Health (Ovid Medline, Biological Abstracts; https://www.wolterskluwer.com/en/solutions/ovid/global-health-30?, accessed on 14 March 2024) with a list of search terms (Table 1) used either individually or in combination as keywords (Figure 1, Step 1). When possible, the search in online databases was directly limited to articles published between January 2000 and January 2020 to support the identification of relevant risk factors that are reflective of current industry practices. All retrieved publications were saved in an EndNote X9 database for further analysis. A primary screening of the articles was conducted based on the relevance of titles, abstracts, and keywords, followed by a secondary screening of records, which involved the exclusion of papers according to the parameters presented in Figure 1 (Step 2). A full reading of the papers was conducted as a third step to identify potential factors or assessment criteria. Moreover, additional factors and assessment criteria were identified by the TWG based on specific processes implemented in the Canadian rendering industry and monitoring and inspection tasks performed by the CFIA (Step 4). A final refinement was applied to retain factors and criteria based on the availability of data sources, the clarity of definitions, the capacity to discriminate risks between plants, and measurability (i.e., an objective assessment can be performed in the context of an audit or inspection process). Factors sharing a similar focus were merged and re-defined or clarified when appropriate. All factors identified were assigned to one of the following clusters, which were defined elsewhere [18]: inherent risk factors; mitigation factors; or compliance factors. The two final steps for the selection of factors (Steps 5 and 6) were conducted from 9 April 2020, to 11 March 2021, through an iterative process involving 5 virtual meetings between the SAC and the TWG. This work resulted in two full rounds of review by the SAC and a finalized list of factors and criteria that were included in the model as of May 2021.

## 3. Results

As depicted in Figure 1, a total of 604 non-duplicate articles were initially found using the search terms listed in Table 1. Two screening procedures were applied, which resulted in the exclusion of 552 publications (Figure 1, Steps 1 and 2). Most of the excluded articles fell outside the scope of the model. For example, many papers were related to edible rendering practices, pet food manufacturing, or nutritional concerns or resulted from ambiguity in the meaning of the verb “to render”. After a full reading of the remaining 52 records, 50 were retained to support the inclusion of feed-safety-related factors and assessment criteria (Figure 1, Step 3). During both rounds of revision by the SAC, exclusions, refinements, or additions were applied to the initial list. This process generated 32 factors, which were classified into three groups (4 inherent, 8 mitigation, and 20 compliance factors, presented in Table 2, Table 3, and Table 4, respectively), selected for inclusion in the ERA-Renderer model. From these, 20 factors originated from the scientific literature, whereas 12 were added based on expert advice and available CFIA inspection data (Figure 1, Steps 4 and 5). The final list of factors was characterized by a total of 179 assessment criteria (Table 2, Table 3 and Table 4). 

## 4. Discussion

From 2013 to 2016, to support the risk-informed oversight of regulated parties, the CFIA developed ERA models to quantify the food safety risk associated with products derived from domestic food establishments and hatcheries [16,17]. Additionally, in 2018, an ERA model was developed to cover both animal and human health risks pertaining to Canadian feed mill products and operations [14]. All of these risk assessment models were developed following a similar approach, which is also being applied for the current ERA-Renderer model. Hence, the present study was conducted with the goal of identifying and selecting feed-safety-related factors to be included in the ERA-Renderer model based on a scientific literature search and expert consultation. 

Due to the wide variety of products related to the rendering sector, challenges were encountered during this study when defining the scope of the literature review. Indeed, rendering facilities collect animal by-product materials, including grease, blood, feathers, offal, and entire animal carcasses, to be processed into rendered products. These products include edible and inedible tallow, lard and greases, feed fats (yellow grease and poultry fat), animal protein meals, hides and skins, and gel bone. Although most of the high-quality inedible fats and proteins are used in the animal feed industry, some inedible rendered products are also directed to the production of essential ingredients for industrial products like pharmaceuticals, lubricants, plastics, biodiesel, or fertilizers, as well as consumer products such as soaps, cosmetics, deodorants, perfumes, and cleaners [6]. Even if the principles of the rendering process remain the same, the health risks associated with finished rendered products vary considerably according to their end use (e.g., for manufacture of livestock feed versus for manufacture of pharmaceutical products). Hence, in this study, the screening of the literature led to the exclusion of many papers enabling only the consideration of feed safety risks, i.e., risks to animal health from the consumption of feed and risks to human health derived from the consumption of products of animal origin, associated with inedible rendering facilities that manufacture feed ingredients intended for livestock. While a total of 50 papers enabled the identification of 20 feed safety factors, 12 additional factors were added based on expert advice and the consideration of available establishment compliance information related to current and upcoming Canadian regulations. In addition, 21 of the 50 papers retained from the literature were specifically linked to the BSE risk, often describing methods to identify animal species in meat and bone meals. Such results most likely reflect the impact of the feed ban enforced in many countries, which directed research efforts towards the prevention and control of the BSE prion in rendering facilities. In fact, BSE is a disease listed by the World Organization for Animal Health (WOAH, founded as OIE) that must be reported by its members in accordance with the organization’s Terrestrial Animal Health Code [69]. However, recent data have shown that other health hazards, such as *Salmonella* spp., polychlorinated biphenyls, and the porcine epidemic diarrhea virus, must also be considered when conducting a feed safety risk assessment of rendered products [36,44,53,70]. In this context, as the scientific literature regarding the general scope of the model was limited, an expert consultation was needed to gather expertise on all types of feed safety risks related to rendering products, operations, and activities in the country. 

### 4.1. Inherent Risk Factors

For the inherent risk factor cluster, most of the scientific literature focused on the type of products used in the rendering process. Indeed, the initial list of feed products extracted from the articles was long and included specific rendered products such as feather meals, organ meals, insect meals, and poultry by-products [21,36,37,52,53,65]. However, based on expert advice, there was a need to focus on the higher-risk products for both animal and human health. In addition to the SRMs and prohibited material already included as BSE-specific risk factors in the current CFIA inspection program for rendering plants, the model covers a wider set of risks by including three additional product categories used for processing in these types of plants (i.e., animal fats and/or oils of restaurant origin, oil and fat products from aquatic animals, and blood and plasma-based products). 

Similar to the ERA-Feed Mill model [14], the importation of feed ingredients was associated with an increase in the risk of disease introduction or drug residues in the food chain, thus raising the feed safety risk associated with rendering plants conducting importation and/or using imported products [38,45,46]. However, some control measures are already applied upstream to feed ingredients purchased from foreign countries, as they can be subject to a pre-market assessment and/or registration process, unless these ingredients already meet the definitions in the list of approved ingredients for manufacture, sale, or import in Canada [71]. In addition, renderers can import some raw ingredients for further processing, but this importation requires specific animal health permits and/or certifications. In some cases, the Canadian feed industry relies heavily on the importation of specific ingredients that are not always available nationally (e.g., antioxidants, preservatives), which limits the discriminatory power of risk between plants when considering such imported products. Hence, it was advised by the expert panel to focus on at-risk practices in plants regarding the cross-contamination risk between domestic and imported rendered products. As such, the mixing of the two types of products during processing or storage was included in the model. 

During the review of factors, the experts suggested adding the type of process related to the use of continuous cooking systems. In Canada, rendering plants usually apply two types of cooking systems: a single continuous in-feed and continuous out-feed cooker and/or several parallel batch cookers that process distinct batches of raw feed materials [6]. Studies identified by the experts that were not primary research studies showed that continuous cooking systems were often not effective at destroying some bacterial species (e.g., *Salmonella* Enteritidis, *Salmonella* Dublin) or prions, whereas batch cooking systems were associated with a higher reduction in pathogen load and infectivity [72,73,74]. Finally, as for all ERA models, the distribution volume of rendered products for each plant was included following expert opinion as a measure of the risk exposure to livestock and, to some extent, human consumers [75]. 

### 4.2. Mitigation Factors

While conducting the literature review, it became apparent that information on specific mitigation measures implemented by the rendering industry to reduce or control their feed safety risk is scarce. Indeed, 15 of 18 assessment criteria included in the mitigation factor cluster originated from expert advice. Most studies supported general good manufacturing practices throughout the rendering process, including the control of incoming supplies, control procedures during processing, the prevention of cross-contamination risks, product sampling, and quality control programs or requirements. It is generally presumed that raw materials, coming from various sources such as butcher shops, supermarkets, restaurants, slaughterhouses, or farms, that enter rendering facilities are at high risk of being contaminated [54,76]. Rendering processes are intended to kill pathogenic microorganisms and to produce high-quality products. While there is evidence to support the notion that rendering procedures generate safe products as a result of the application of high temperatures for suitable periods, contaminated finished rendered products are still reported, but the sources of contamination are often unclear [77]. Indeed, in a recent review, Vidyarthi et al. [55] reported the need for research studies tackling the sources of microbial contamination in rendering plants. Many studies have raised the possibility of the sustained contamination of products following inadequate rendering processes or post-processing contamination events involving rendered products and rendering equipment, the environment, or transportation [78,79]. In this context, the mitigation measures to be included in the ERA-Renderer model are mostly related to the control of rendering procedures and the prevention of post-processing contamination of finished rendered products. 

### 4.3. Compliance Risk Factors

For the purposes of the ERA-Renderer model, compliance factors were identified based on the rendering plant’s historical and upcoming inspection data under the CFIA’s Integrated Agency Inspection model [80], which will be implemented once the modernized Feeds Regulations come into force. Indeed, with the upcoming regulations [81], the development, implementation, and update of preventive control plans (PCPs) will be required to achieve sustainable feed safety standards in all types of feed establishments, including rendering plants. 

Along with the support of the scientific literature, the risk factors and criteria included in the compliance cluster of the model are all covered in the latest version of the Food and Agriculture Organization (FAO) manual for implementing the Codex Alimentarius Code of Practice on Good Animal Feeding [82]. In this manual, the FAO suggests that competent authorities account for factors such as the history of the conformity of regulated parties when designing the frequency and intensity of controls in inspection systems. The compliance factors included in the ERA-Renderer model are well aligned with this recommendation by considering the current and historical timeframes defined by experts for each risk factor according to the impact on the risk assessment (i.e., current and past three inspection results up to the last three years for each PCP, enforcement actions in the past two years, animal and human health complaints in the past year, and feed safety recalls in the past 5 years). Historical compliance results may help reflect the company’s management commitment to feed safety, which should ensure the effectiveness of the feed safety system in place. While the model recognizes the risk mitigation efforts of the industry, the compliance portion of the model is expected to reflect the appropriateness of their implementation by rendering facilities and, thus, promote compliance. Interestingly, the compliance cluster of all the CFIA’s ERA models contains almost the exact same risk assessment criteria [14,16,18], which is related to the upcoming enforcement of similar regulatory requirements (e.g., licences, PCPs, traceability requirements) and the implementation of a standardized inspection process. This common compliance assessment will allow for future inter-sector comparisons by the agency.

It is widely recognized that the safety of the food chain relies on a farm-to-fork approach, covering pre-harvest, harvest, and post-harvest sectors [83]. This type of approach supports the need to consider the health of plants, animals, and humans, creating a One Health framework to ensure food safety. The development of the ERA-Renderer model is one of the very few tools covering both animal and human health risk factors, as conducted for the CFIA’s ERA-Feed Mill model [14]. These risk assessment models offer an innovative evaluation of combined health risks in pre-harvest sectors of the food chain, which has previously been conducted through separate health risk assessment frameworks [84]. Risk assessment is known to be a complicated task, comprising hurdles such as lack of evidence, the need for high-level skills and knowledge of different disciplines, and the need for field studies testing the applicability of the approach and clear, effective communications to ensure proper risk management [84]. Such challenges are greater when integrating multiple health hazards related to feed safety concerns. Indeed, although food safety risk assessments include hazards that have been well studied and that can be quantified for their human health impacts based on a defined metric [85], animal health risk assessments have been conducted in recent years, but no official quantification has been carried out, with studies relying on subject matter expertise elicitations [15,86,87]. Thus, there was a need to gather multidisciplinary high-level expertise from various sectors in order to select feed-safety-related factors and assessment criteria to be included in the ERA-Renderer model. It is important to note that this model is intended to be adaptable according to new scientific data as well as changes within the rendering industry. As such, the list of selected risk factors and assessment criteria will need to be reviewed over time to add any new relevant feed safety risk factor for rendering plants in Canada and/or to remove outdated risk factors following approval by the SAC. For now, the next step in designing the mathematical algorithm of this model will rely on an expert consultation to quantify the relative risk of each assessment criterion on the feed safety risk of rendering plants. 

## 5. Conclusions

The present study enabled the selection of 32 feed-safety-related factors and 179 assessment criteria that are quantitatively measurable, auditable, and discriminant and for which data are available for inclusion in the CFIA’s ERA-Renderer model. This model will be used to categorize rendering plants according to their feed safety risk. By using an evidence-based and consistent approach, the agency will improve its ability to prioritize and allocate inspection resources by targeting rendering plants representing the greatest risks to both animal and human health. With the development of ERA models covering pre-harvest and post-harvest regulated sectors, Canada is becoming more closely aligned with international standards that are transitioning towards a risk-informed approach covering the entire food supply continuum.

## Figures and Tables

**Figure 1 animals-14-01020-f001:**
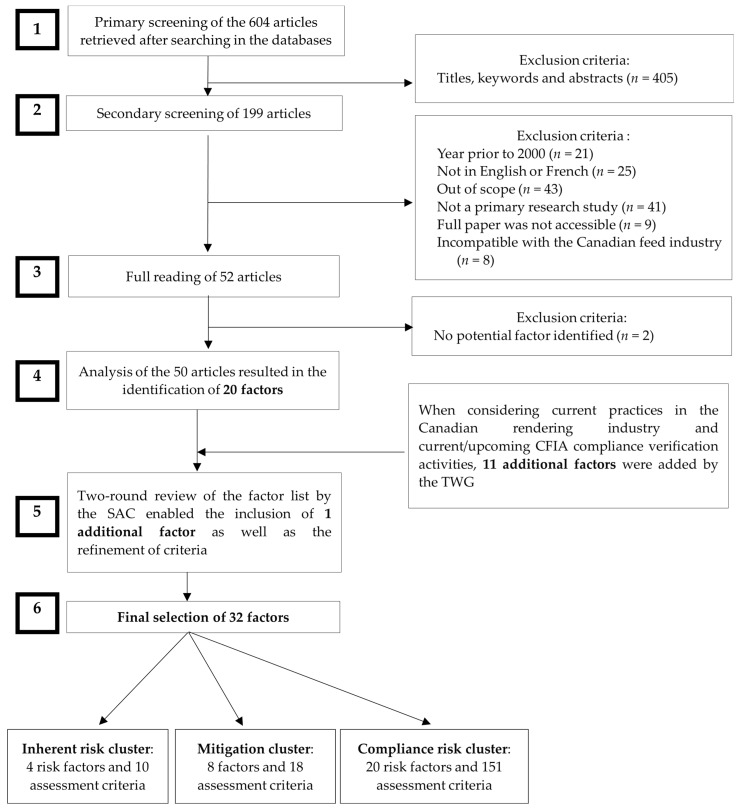
Literature search strategy flow and screening process. SAC; scientific advisory committee. TWG: technical working group.

**Table 1 animals-14-01020-t001:** Combinations of keywords used for the literature search.

Category	Keywords
Health	Animal health, human health, zoonosis, foodborne disease, risk assessment
Feed	Animal feed, feed, rendering, feed safety, food safety
Inherent risk factors	Rendering plants, rendering products, processing steps, prohibited material, biological hazards, chemical hazards, specified risk materials, animal protein products, meat and bone meal, blood meal, feather meal, tallow, animal by-products
Mitigation factors	Irradiation, thermal processing, heat treatment, good manufacturing practices, audits, suppliers’ control, chemical additives, chemical treatment, HACCP, sampling plan, process controls, quality assurance program, prevention, mitigation, hazard mitigation, certification, preventive control plan, risk mitigation, risk management, cross-contamination controls
Compliance risk factors	Regulation, traceability, complaints, enforcement action, sampling results, product recall, biosecurity, recall plan, track records, equipment, sanitation, pest management, transportation, storage, employee hygiene, feed hygiene, building maintenance, employee training, inspection, compliance

**Table 2 animals-14-01020-t002:** List of inherent risk factors and their related assessment criteria selected for inclusion in the ERA-Renderer model. Risk factors and criteria identified from the scientific literature are followed by their relevant references in parentheses, whereas those identified based on expert advice are listed in italics.

Inherent Risk Factor	Assessment Criteria
*Annual distribution volume*	*Contribution to the Canadian distribution volume*
*Type of process*	*Continuous cooking system*
Type of products used for processing or manufacture	Use of animal fats and/or oils of restaurant origin as incoming materials [19] Handling of specified risk materials (SRMs) [20] ⚬ Exclusive reception of non-SRMs ⚬ Reception of both SRMs and non-SRMs Manufacture of prohibited material (PM) [21,22,23,24,25,26,27,28,29,30,31,32,33,34,35] ⚬ Exclusive manufacture of products not containing PM ⚬ Manufacture of both products containing PM and products not containing PM Manufacture of oil and fat products from aquatic animals [19,36,37,38,39,40,41,42,43] Manufacture of blood and plasma-based products [34,44]
Handling of imported finished rendered products [38,45,46]	*Absence of segregation during storage and/or mixing of domestic and imported products*

**Table 3 animals-14-01020-t003:** List of mitigation factors and their related assessment criteria selected for inclusion in the ERA-Renderer model. Factors and criteria identified from the scientific literature are followed by their relevant references in parentheses, whereas those identified based on expert advice are listed in italics.

Mitigation Factor	Assessment Criteria
Feed safety control measures for incoming supplies [38,43,44,45,47]	*Supplier contract letters* *Letters of guarantee* *Certificates of analysis* *On-site audits of suppliers*
Feed safety control measures during processing [20,24,29,44,48,49,50,51]	*Use of processing methods for animal proteins recognized by the European Union* *Use of traceable markers in continuous cooking systems* ⚬ *To validate rendering processes (e.g., TiO2, rice DNA)* ⚬ *To monitor rendering efficacy (i.e., routinely used) using an approved product* Use of thermocouples to control cold-spot locations [52]
Feed safety control measures for preventing cross-contamination [48,53,54,55,56]	*Complete segregation between raw and finished rendered products*
Feed safety control measures for transportation [29,55]	*Using or requiring dedicated conveyance for specific incoming materials* *Using or requiring dedicated conveyance for specific types of finished rendered products*
Sampling of finished rendered products (for feed safety)	Authentication tests of finished rendered products targeting meat and bone meals [24,26,27,32,33,57,58,59,60,61,62,63] Hazard sampling program for finished rendered products [29,64] ⚬ *Hazard sampling plan in place that includes analysis and corrective actions* ⚬ *Hazard sampling plan in place and trend analysis performed* ⚬ *Hazard sampling plan in place, trend analysis performed, and systemic actions taken if unusual trend is detected*
*Feed safety assurance personnel*	*Having at least one employee dedicated to and responsible for feed safety assurance who is available full-time during production hours*
*Feed safety certifications*	*HACCP-type certification (e.g., Safe Feed/Safe Food * for renderers from the American Feed Industry Association, FeedAssure^®^ **)*
*Third-party audits*	*Being audited by a third party for feed safety, other than the audits linked to a certification (e.g., client audit)*

* The feed/food safety management and certification program (FSC36) developed by the American Feed Industry Association (AFIA), designed for the American and Canadian feed industries. ** The Hazard Analysis Critical Control Point (HACCP) program developed by the Animal Nutrition Association of Canada (ANAC) for the Canadian feed industry.

**Table 4 animals-14-01020-t004:** The list of compliance risk factors, their respective timeframes considered by the model, and their related assessment criteria selected for inclusion in the ERA-Renderer model. Risk factors and criteria identified from the scientific literature are followed by their relevant references in parentheses, whereas those identified based on expert advice are listed in italics. Note that for each government inspector assessment of the establishment’s preventive control plan (PCP) program, the model includes the number of non-compliances with a direct impact on feed safety for the current and last three inspections and the number of non-compliances with a potential impact on feed safety for the current and last three inspections (i.e., a total of 8 assessment criteria for each PCP program).

Compliance Risk Factor		Assessment Criteria
Assessment of the establishment’s preventive control plan (current and last three inspection results, up to the last three years)	Incoming products program [36,38,43,44,45,47]	*Presence of non-compliance related to the incoming products program as assessed by a government inspector (current and past three inspection results)*
	Process control program [20,24,25,27,29,30,44,49,50,51,52,53]	*Presence of non-compliance related to the process control program as assessed by a government inspector (current and past three inspection results)*
	End-product control program [20,22,24,25,26,27,29,31,32,33,34,40,41,57,58,59,60,61,62,63,65,66,67]	*Presence of non-compliance related to the end-product control program as assessed by a government inspector (current and past three inspection results)*
	Import control program [36,38,45,46]	*Presence of non-compliance related to the import control program as assessed by a government inspector (current and past three inspection results)*
	Sanitation, biosecurity, and biocontainment program [21,53]	*Presence of non-compliance related to the sanitation, biosecurity, and biocontainment program as assessed by a government inspector (current and past three inspection results)*
	Pest control program [53]	*Presence of non-compliance related to the pest control program as assessed by a government inspector (current and past three inspection results)*
	Hygiene and biosecurity program [53]	*Presence of non-compliance related to the hygiene and biosecurity program as assessed by a government inspector (current and past three inspection results)*
	Employee training program [29]	*Presence of non-compliance related to the employee training program as assessed by a government inspector (current and past three inspection results)*
	Equipment design and maintenance program [53]	*Presence of non-compliance related to the equipment design and maintenance program as assessed by a government inspector (current and past three inspection results)*
	Premises and surroundings program [53]	*Presence of non-compliance related to the premises and surroundings program as assessed by a government inspector (current and past three inspection results)*
	Buildings program [53]	*Presence of non-compliance related to the buildings program as assessed by a government inspector (current and past three inspection results)*
	Receiving, transportation, and storage program [21,29,32,44,53,68]	*Presence of non-compliance related to the receiving, transportation, and storage program as assessed by a government inspector (current and past three inspection results)*
	Traceability and control program [65]	*Presence of non-compliance related to the traceability and control program as assessed by a government inspector (current and past three inspection results)*
	*Chemicals program*	*Presence of non-compliance related to the chemicals program as assessed by a government inspector (current and past three inspection results)*
	*Water, ice, and steam program*	*Presence of non-compliance related to the water, ice, and steam program as assessed by a government inspector (current and past three inspection results)*
	*Waste disposal program*	*Presence of non-compliance related to the waste disposal program as assessed by a government inspector (current and past three inspection results)*
	*Complaints program*	*Presence of non-compliance related to the complaints program as assessed by a government inspector (current and past three inspection results)*
*History of enforcement actions or control measures (last 2 years)*		*History of enforcement actions or control measures taken by the regulatory agency in the past two years:* ⚬ *Seizure and detention of products;* ⚬ *Removal of products imported into Canada;* ⚬ *Final notice of non-compliance;* ⚬ *Meeting with regulated party;* ⚬ *Notice of violation with warning;* ⚬ *Notice of violation with penalty;* ⚬ *Permit suspension;* ⚬ *Permit cancellation;* ⚬ *Refusal to renew a permit;* ⚬ *Prosecutions;* ⚬ *Issuance of a mandatory recall order.*
*Confirmed animal and human health complaints (past year)*		*History of confirmed animal and/or human health complaints taken by the regulatory agency in the past year:* ⚬ *1–4 confirmed animal and/or human health complaints;* ⚬ *≥5 confirmed animal and/or human health complaints.*
*Feed safety recalls (last 5 years)*		*History of feed safety recalls taken by the regulatory agency in the past five years: * ⚬ *1 feed recall;* ⚬ *≥2 feed recalls. *

## Data Availability

Data are contained within the article.

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
