# Peer review of "The Development of a Risk Assessment Model for Inedible Rendering Plants in Canada: Identifying and Selecting Feed Safety-Related Factors"

_animals, 2024, doi:10.3390/ani14071020_

Round 1

Reviewer 1 Report

Comments and Suggestions for Authors

Congratulations to the authors for a well-written manuscript describing the risk assessment model for rendering plants in Canada with the manuscript objective of reporting the initial steps of the model to identify and select animal feed and human food safety-related factors. You clearly defined the literature search criterion and limitations as well as the need to have additional input from invited experts to help identify and select other factors and assessment criteria. You also described that your long-term goal of the model was to help determine allocation of inspection resources for rendering plants based on the eventual final model established risks.  

I have only a few minor comments or suggestions. 

Line 26:  delete the space between the words "select  factors"

Line 146 and Figure 2 exclusion criteria: Why not include results of studies published prior to the year 2000? I note you excluded 21 studies, however if these studies provided appropriate criteria, why not use them? It seems you just arbitrarily dropped out studies published before 2000? I suggest some justification statement around line 146 related to why year 2000 was used as a limitation of the search. Also, I understand you limited the literature search to May 2021, but this is now 2024 and perhaps there is more information available since 2021. I suggest you add a statement that as the next steps of the model development progress an updated search of literature will be done to ensure no other criterion related to inherent risks, mitigation factors or compliance factors have been missed or overlooked. 

Again, congratulations and I look forward to future publications related to further development of this model for assessing risks and resources needed to assure feed and food safety of rendered animal products. Rendering is a very important recycling service that is essential to maintain sustainable animal agriculture and help reduce the carbon footprint. 

Reviewer 2 Report

Comments and Suggestions for Authors

Manuscript animals-2880546, entitled “Development of a risk assessment model for inedible rendering plants in Canada: identifying and selecting feed safety-related factors

This article provides useful information addressing the development of a risk assessment model for inedible rendering plants in Canada. It is in general appropriately organized, carried out and written, however there are some points that should be corrected or clarified.

Authors develop a Hazard Analysis and Critical Control Points (HACCP) system adjusted to inedible rendering plants and describe its first stages. Although, it is easy to follow the 32 feed safety-related factors, it is difficult for the 179 assessment criteria. Could you please assign a number next  to them (from 1 to 179) in Tables 2-4?

L43: “developed” instead of “produced”

L53-54: Please use the same measurement unit (product quantity) for cattle, hogs, poultry and fish.

L161: Please convert pounds to kg (or tn)

L81: What do you mean by “in addition to food”?

L92: “for the supervision of the whole procedure” instead of “to oversight”

L98, 110: “permission” instead of “permit”

L157: “performed” instead of “done”

L181: “classified” instead of “divided”

L222: “Due to” instead of “Given”

L262: “covers” instead of “will cover”

L308: “…et al. [39] reported in a…”

L313: “…model are mostly related to the control..”

L360: “carried out” instead of “done”

Comments on the Quality of English Language

Minor editing of English language required
